# People who inject drugs in metropolitan Chicago: A meta-analysis of data from 1997-2017 to inform interventions and computational modeling toward hepatitis C microelimination

**Basmattee Boodram**[1☯], **Mary Ellen Mackesy-Amiti**[1☯]*, **Aditya Khanna**[2], **Bryan Brickman**[3], **Harel Dahari**[4], **Jonathan Ozik**[5]

1 Division of Community Health Sciences, School of Public Health, University of Illinois at Chicago, Chicago, Illinois, United States of America, 2 Department of Behavioral and Social Sciences, School of Public Health, Brown University, Providence, Rhode Island, United States of America, 3 Department of Medicine, Chicago Center for HIV Elimination, University of Chicago, Chicago, Illinois, United States of America, 4 Division of Hepatology, Department of Medicine, Loyola University Medical Center, Maywood, Illinois, United States of America, 5 Decision and Infrastructure Sciences Division, Argonne National Laboratory, Lemont, Illinois, United States of America

☯ These authors contributed equally to this work.
* mmamiti@uic.edu

**Data Availability Statement:** All data and documentation files are available from the Open

## Abstract

Progress toward hepatitis C virus (HCV) elimination in the United States is not on track to meet targets set by the World Health Organization, as the opioid crisis continues to drive both injection drug use and increasing HCV incidence. A pragmatic approach to achieving this is using a microelimination approach of focusing on high-risk populations such as people who inject drugs (PWID). Computational models are useful in understanding the complex interplay of individual, social, and structural level factors that might alter HCV incidence, prevalence, transmission, and treatment uptake to achieve HCV microelimination. However, these models need to be informed with realistic sociodemographic, risk behavior and network estimates on PWID. We conducted a meta-analysis of research studies spanning 20 years of research and interventions with PWID in metropolitan Chicago to produce parameters for a synthetic population for realistic computational models (e.g., agent-based models). We then fit an exponential random graph model (ERGM) using the network estimates from the meta-analysis in order to develop the network component of the synthetic population.

## Introduction

Hepatitis C virus (HCV) infection is a leading cause of chronic liver disease and mortality worldwide. Globally, people who inject drugs (PWID) are disproportionally represented

Science Framework database (DOI 10.17605/OSF.IO/4PXZW).

**Funding:** This work was supported by grants to HD, JO and BB[1] from the National Institute of General Medical Sciences, USA (https://www.nigms.nih.gov), grant number R01GM121600 and to BB[1] from the National Institute of Drug Abuse, USA (https://www.drugabuse.gov), grant number R01DA043484. The contents of this paper are solely the responsibility of the authors and do not necessarily represent the official views of NIGMS or the National Institutes of Health. The funders had no role in study design, data collection and analysis, decision to publish, or preparation of the manuscript.

**Competing interests:** The authors have declared that no competing interests exist.

among the estimated 71.1 million chronically infected globally [1]. In the United States (U.S.), PWID are at the highest risk for acquiring and transmitting HCV infection primarily through contaminated syringe-sharing [2]. Data from a recent modeling study that included 88 countries (85% of the global population) showed that if transmission risk among PWID was removed, an estimated 43% of incident HCV infections would be prevented from 2018–2030 [3]. Despite the long-term availability of harm reduction strategies such as syringe services programs (SSPs), medication-assisted therapy (MAT), behavioral counseling, and the recent availability of highly-effective direct-acting antiviral (DAA) treatment, injection drug use (IDU) and HCV incidence have been increasing at alarming rates among PWID, particularly among young, non-Hispanic white, nonurban PWID [4–9].

Currently, the U.S. lags behind other high-income countries in progress towards achieving the World Health Organization's (WHO) goal of reducing new chronic infections by 90% and mortality by 65% by 2030 [10]. The HCV microelimination approach [11], which entails pursuing elimination goals in discrete high-risk populations such as PWID and/or in specific regions, may be a more pragmatic, less daunting approach to ultimately achieving WHO's global HCV elimination goal. However, even within the U.S., the PWID population is heterogeneous, as evidenced by subpopulation and geographic differences in HCV incidence and prevalence [9, 12, 13]. Moreover, significant barriers to DAA treatment initiation and completion among PWID exist [14]. An understanding of the complex interplay of individual, social, and structural level factors that might alter HCV incidence, transmission, and HCV treatment uptake among PWID is imperative for achieving HCV microelimination in this population. While this level of complexity is extremely difficult to address with traditional empirical studies, it could be theoretically investigated using an individual oriented computational modeling approach such as agent-based modeling (ABM) [15–17].

We report results of a meta-analysis of recent studies (1997–2017) [5, 18–27] among PWID to profile this population using data representing the estimated 32,000 PWID who reside in metropolitan Chicago, Illinois [28] that includes both urban and suburban PWID [29]. Our findings could be used to 1) better inform development and implementation of interventions targeting PWID subpopulations, and 2) to provide a realistic profile of a synthetic PWID population for computational models that could be used to forecast HCV epidemiology and discover the specific combination of interventions that would achieve HCV microelimination. To further assist with the development of a realistic synthetic population, we also fitted an exponential random graph model (ERGM) using the network estimates from the meta-analysis. ERGMs offer a uniquely robust and flexible approach in modeling the dependencies between behavioral parameters, and have been used to model syringe sharing networks in a number of related contexts [30].

## Methods

### Meta-analysis data sources

We used twelve diverse datasets in the meta-analysis, including eleven research studies and one SSP (Table 1), which were selected on the basis that they would provide rich data sources on the highly heterogeneous PWID population of metropolitan Chicago. The SSP data were split into two datasets according to year of enrollment to represent earlier and later time periods. The primary drug injected among PWID in the metropolitan Chicago area is heroin, with cocaine second. Methamphetamine injection was uncommon during the covered time period. In all datasets, records only included individuals 18 years or older and excluded those missing key demographic information (sex, age, race/ethnicity).

**Table 1. Data sets included in the PWID meta-analysis.**

| Study | Years | Sample size | Age range | Reference |
|---|---|---|---|---|
| CIDUS 2 | 1997–1999 | 695 | 18–34 | [18] |
| CIDUS 3 | 2002–2004 | 775 | 18–34 | [19] |
| NATHCV | 2002–2005 | 110 | 18–34 | [20] |
| NIHU | 2004–2006 | 150 | 18–34 | [21] |
| SATHCAP | 2005–2009 | 1356 | 18 + | [22] |
| PSYCH | 2008–2010 | 610 | 18–25 | [23] |
| NHBS IDU2 | 2009 | 553 | 18 + | [24] |
| NHBS IDU3 | 2012 | 210 | 18 + | [25] |
| SOCNET | 2012–2013 | 164 | 18–34 | [5] |
| NHBS IDU4 | 2015 | 541 | 18 + | [26] |
| MOOD | 2016–2017 | 180 | 18–34 | [27] |
| COIP SEP (1) | 2005–2010 | 4952 | 18 + | |
| COIP SEP (2) | 2011–2016 | 2632 | 18 + | |

PWID: People who inject drugs

CIDUS: Collaborative Injection Drug Users Study (CDC)

NATHCV: Early natural history of HCV infection among injection drug users (NIDA)

NIHU: Non-injected heroin use, HIV, and transitions to injection (NIDA)

SATHCAP: Sexual Acquisition and Transmission of HIV Cooperative Agreement Project (NIDA)

PSYCH: Psychiatric disorders and HIV risk behaviors among young injection drug users (NIDA)

SOCNET: Social networks of young injection drug users (Chicago DCFAR)

MOOD: Emotion dysregulation and risky behavior among people who inject drugs (NIDA)

NHBS: National HIV Behavioral Surveillance (CDC).

Sexual orientation was only collected in 9 of the 11 studies, and transgender identity was collected in all studies. Subjects who identified as transgender were excluded if the study did not include a variable to classify them as male or female (i.e., biological sex). The datasets were stratified by sex (male or female), race/ethnicity (non-Hispanic White, non-Hispanic Black, Hispanic, and Other/mixed), and age category (18–24, 25–34, 35–44, 45 and older). Measures were harmonized across studies, and frequencies of categorical variables and means and standard deviations of continuous variables were computed for the strata in each dataset. Poisson standard deviations were computed for count variables such as network size and number of sex partners.

## Measures

**Sociodemographic characteristics.** Measures included basic demographic categories of sex, age, race/ethnicity, and place of residence (dichotomized as within the city limits of Chicago (urban) vs. all other surrounding suburban areas). Measures of current employment, income, and recent homelessness were also available. Several studies asked about sources of income, while others asked directly about employment status. The measures were harmonized by creating indicators of any employment (including temporary work), and regular employment (full or part-time). Measures of income were answered on a categorical scale, in increments of $500 or $5000. We used the median of the category (the lower limit for the top category) as the value of income. Homelessness was measured as self-perceived homelessness (e.g. have you been homeless, have you considered yourself homeless) within the past 6 or 12 months.

**Harm reduction.**   We included measures of current substance use treatment (other than peer support groups), and obtaining syringes from a SSP in recent months.

**IDU networks.**   Two measures of IDU network size were available: 1) number of people respondent knows who inject drugs (PWID network, 6 studies); 2) number of people respondent injected drugs with in the past 30 days, 3 months, or 6 months (injection network, 5 studies). Three studies included both of these measures. The syringe-sharing network is a further subset of the network of people with whom they injected. We estimated the number of people respondent used a syringe after (in-degree), and the number of people respondent gave his or her syringe to after using it (out-degree). As measures of network mixing, we estimated the network proportions by sex, race/ethnicity, and Chicago/non-Chicago residence, based on responses to questions included in studies that used respondent driven sampling (RDS). However, only one study (SATHCAP) included respondents of all ages, therefore for purposes of informing the model, network mixing estimates were based on this study only.

**IDU behaviors.**   Variables related to injection drug use included age of first injection drug use, years of injecting (computed from current age and age of first injection), and frequency of injection in past 30 days. Frequency of injection was estimated based on responses to two questions: 1) average frequency of injection in the past 3, 6 or 12 months, with categorical responses, from once a month or less to every day, and 2) typical number of times injected per day, from once to 10 or more times a day.

Injection risk behaviors included syringe-mediated drug sharing (SMS) in the past 3, 6, or 12 months (injected drugs with a syringe after someone else squirted drugs into it from another syringe, "backloading", or drugs were mixed, measured, or divided using someone else's syringe), any receptive syringe sharing (RSS; injected with a syringe that someone else had used to inject) in the past 30 days to 12 months, frequency of RSS (estimated proportion of injections that involved RSS based on Likert scale response or number of RSS injections and frequency of injection), any equipment sharing (shared cookers, cotton, or rinse water) in the past 30 days to 12 months, and frequency of sharing cookers (converted Likert scale responses to proportion of injections).

## Meta-analysis: Random effects model

Under the fixed-effect model it is assumed that the true effect size for all studies is identical, and the only reason the effect size varies between studies is due to sampling error. By contrast, under the random-effects model the goal is not to estimate one true effect, but to estimate the mean of a distribution of effects [31]. Although our studies have many similarities, there may be important differences between the study populations that influence the observed effect sizes, including most obviously time. There is also variability in the measures used across the studies that can affect effect sizes, e.g. reporting on behavior in the past 3 months vs. past 6 months. Moreover, due to confounding of study with time, we did not perform analyses of time trends. In another paper, using meta-regression on aggregate level data [32], we report an increase in homelessness over time among young PWID. We detected no significant changes in risk behavior or networks over time.

Estimates of network size and injection risk behavior were stratified by sex, race and ethnicity (non-Hispanic white, non-Hispanic Black, Hispanic (all races), and non-Hispanic other races), and age category (18–24, 25–34, 35–44, and 45 and older.) Random effects meta-analysis of proportions and means was conducted in R using the meta package [33]. Meta-analysis of continuous and count variables was conducted using the inverse variance method of DerSimonian and Laird [34] and the GLMM model was used for proportions. The REML estimator was applied to estimate between study variance (tau-squared) [35, 36]. Poisson standard

deviations were used in the estimation of count variables (network size, number of people sharing syringes and other equipment). The distribution of random effects was assumed normal. Sample estimates based on subgroups of fewer than five subjects were excluded.

### Mixed effects regression analyses

We estimated mixed effects regression models in Stata (v. 15, StataCorp) to examine main effects of demographic variables on socioeconomic status, harm reduction, networks, and risk behavior. Logistic regression models were estimated for binary outcomes, and negative binomial regression models were estimated for count outcomes. Contrasts were computed to test the joint effects of multi-category variables of race/ethnicity and age category. We conducted separate analyses to examine associations with sexual orientation using data from those studies that collected this information.

### Modeling networks of persons who inject drugs

The theoretical framework for modeling IDU networks is based upon the exponential random graph models (ERGMs) [37–40]. The ERGMs are fit using the ergm package [38, 41] in the R programming language [42] to generate directed graphs that represent the syringe sharing network. Conceptually, ERGMs can be understood to be similar to logistic regression models, where the outcome variable indicates presence and absence of an "edge" (also known as "tie" or "relational information") and the independent variables describe network configurations.

Because PWID syringe-sharing relationships are not random [43], it is necessary to include mixing parameters in the model that govern the probability of tie formation. Characteristics such as sex, age, race and ethnicity, and geographic proximity are important factors that influence both tie formation and risk behaviors related to HCV transmission and acquisition [43–46].

The log-odds of formation of each partnership type were dependent upon the number and distribution of existing syringe-sharing relationships (henceforth, "relationships") within the network. The mean number of such relationships was estimated from meta-analyses as described above, based on reported numbers of receptive and distributive syringe sharing partners, and the distribution of relationships was determined by the following parameters: mixing based on sex ("male" and "female"), race/ethnicity ("non-Hispanic white", "non-Hispanic Black", "Hispanic", "non-Hispanic other") and age ("under 25" vs. "25 and older"); the distribution of geographic distances and the distribution of in- and out-degree edges. The mathematical formulation of the model is given in Equation A.1 in the S1 File.

Standard tools in the statnet package were used to assess model fit and convergence. One hundred networks were simulated from the fitted model and the distributions of the simulated parameters were compared to the targets estimated from the meta-analysis above.

## Results

### Meta-analysis estimates

Heterogeneity among studies was high for nearly all measures, with I-squared > 90%. Selected characteristics of the combined samples are shown in Table 2. Additional output is available in S1 Table.

**Socioeconomic status.** On average 43% (95% CI 0.26–0.63, tau = 1.26) of PWID reported having some kind of employment including temporary work, and 19% (95% CI 0.09–0.36, tau = 1.37) of PWID were employed in a regular job (full or part-time). Women were less likely to be employed in any capacity (OR = 0.49, 95% CI 0.36–0.65), as were Hispanic (vs.

**Table 2. Estimated characteristics of Chicago metro area PWID, random effects meta-analysis of proportions.**

| Parameter | # studies | Estimate | 95% Conf. Int. | | tau | $I^2$ |
|---|---|---|---|---|---|---|
| male | 13 | 0.68 | 0.659 | 0.698 | 0.128 | 0.718 |
| non-Hispanic white | 13 | 0.53 | 0.433 | 0.629 | 0.724 | 0.986 |
| non-Hispanic Black | 13 | 0.14 | 0.076 | 0.234 | 1.182 | 0.982 |
| Hispanic | 13 | 0.23 | 0.200 | 0.268 | 0.323 | 0.910 |
| other race (non-Hispanic) | 13 | 0.04 | 0.026 | 0.049 | 0.526 | 0.831 |
| heterosexual | 9 | 0.85 | 0.807 | 0.884 | 0.428 | 0.971 |
| gay/homosexual | 9 | 0.03 | 0.018 | 0.047 | 0.692 | 0.925 |
| bisexual | 9 | 0.11 | 0.074 | 0.154 | 0.604 | 0.977 |
| MSM | 12 | 0.05 | 0.035 | 0.080 | 0.708 | 0.958 |
| injected heroin by itself | 13 | 0.98 | 0.954 | 0.987 | 1.069 | 0.966 |
| injected cocaine | 10 | 0.36 | 0.307 | 0.406 | 0.329 | 0.927 |
| injected "speedball" | 11 | 0.24 | 0.185 | 0.301 | 0.522 | 0.943 |
| injected methamphetamine | 13 | 0.03 | 0.018 | 0.040 | 0.675 | 0.904 |

PWID: people who inject drugs; MSM: men who have sex with men; SSP: syringe service program; "speedball": heroin and cocaine injected together.

non-Hispanic white, OR = 0.67, 95% CI 0.54–0.84) and non-Hispanic Black persons (vs. non-Hispanic white, OR = 0.63, 95% CI 0.42–0.93), and persons over 25 years old (Chi2[1] = 48.25, p < .0001). Results were similar for regular employment except that Chicago residence decreased the likelihood of regular employment (OR = 0.76, 95% CI 0.61–0.94). The average monthly income was $1,367 (95% CI 1132–1603, tau = 286), with higher income reported by non-Hispanic white PWID (Chi2[1] = 12.84, p = .0003), and decreasing with increasing age (Chi2[3] = 126.91, p < .0001).

An estimated 38% of PWID reported being homeless; those who resided in Chicago were more likely to report being homeless (OR = 1.42, 95% CI 1.16–1.75), while non-Hispanic Black PWID were less likely to be homeless (35% vs. 44% non-Hispanic white, OR = 0.68, 95% CI 0.56–0.83). Sexual orientation was not associated with regular employment, but bisexual men were less likely to have income from any kind of work compared to heterosexual men (OR = 0.75, 95% CI 0.60–0.94). Bisexual men (vs. heterosexual men, OR = 1.69, 95% CI 1.44–1.98), homosexual women (vs. heterosexual women, OR = 2.28, 95% CI 1.40–3.72), and bisexual women (vs. heterosexual women, OR = 1.55, 95% CI 1.23–1.94) were more likely to report being homeless.

**Injection behavior and harm reduction.** The estimated average age of first injection was 22 (95% CI 20.9–23.2, tau = 2.1), duration of injection was 9.9 years (95% CI 6.6–13.1, tau = 5.7), and average frequency of injection in the past 30 days was 75 times (95% CI 60.5–89.8, tau = 26). Over half of PWID (59%, 95% CI 0.59–0.46, tau = 0.84) reported use of a SSP, and 13% (95% CI 0.10–0.17, tau = 0.48) reported current substance use treatment. Women (OR = 1.23, 95% CI 1.04–1.46) and PWID over 25 (Chi2[1] = 10.07, p = .0015) were more likely to use a SSP, while non-Hispanic Black PWID were less likely than others to use a SSP after adjusting for age (Chi2[3] = 22.80, p < .0001). PWID who resided in Chicago were marginally more likely to report using a SSP (OR = 1.28, 95% CI 1.00–1.64). Women were more likely than men to report current treatment (OR = 1.42, 95% CI 1.25–1.61), and current treatment increased with increasing age (Chi2[3] = 424.21, p < .0001). Sexual orientation was not associated with SSP use, but bisexual men were more likely to report current treatment (vs. heterosexual men, OR = 1.36, 95% CI 1.08–1.71); the prevalence of current treatment for bisexual men was similar to that for women.

**Table 3. Meta-analysis estimates of PWID network size, by sex, race or ethnicity, and age category.**

| Sex | Age category | non-Hispanic white | | | non-Hispanic Black | | | Hispanic | | | Other non-Hispanic | | |
|---|---|---|---|---|---|---|---|---|---|---|---|---|---|
| | | Mean | LL | UL | Mean | LL | UL | Mean | LL | UL | Mean | LL | UL |
| Male | 18–24 | 9.81 | 6.32 | 13.31 | 5.13† | - - | - - | 7.24 | 6.06 | 8.43 | 7.33 | - - | - - |
| | 25–34 | 15.77 | 7.56 | 23.97 | 13.44 | 11.97 | 14.90 | 15.26 | 9.20 | 21.33 | 4.39 | 3.20 | 5.57 |
| | 35–44 | 22.19 | 16.11 | 28.26 | 22.53 | 13.05 | 32.01 | 21.66 | 16.22 | 27.10 | 31.83† | | |
| | 45+ | 21.08 | 16.07 | 26.09 | 22.23 | 20.37 | 24.10 | 28.02 | 20.49 | 35.55 | 16.64 | 9.64 | 23.64 |
| Female | 18–25 | 10.85 | 5.01 | 16.69 | 34.00† | - - | - - | 5.85 | 3.09 | 8.61 | 8.12 | - - | - - |
| | 25–34 | 16.51 | 9.79 | 23.23 | NA | - - | - - | 17.51 | 8.59 | 26.43 | 67.5† | - - | - - |
| | 35–44 | 23.18 | 15.69 | 30.67 | 17.62 | 7.05 | 28.19 | 23.93 | 16.01 | 31.84 | NA | - - | - - |
| | 45+ | 20.86 | 17.08 | 24.65 | 21.77 | 13.42 | 30.12 | 28.08 | 18.09 | 38.07 | NA | - - | - - |

LL: lower limit 95% confidence interval; UL: upper limit 95% confidence interval; NA: estimate not available, insufficient data.

† N < 10.

Note: confidence intervals not computed for estimates based on a single study.

**Network size.** Estimates of PWID and injection network size are shown in Tables 3 and 4. PWID network size increased with age (Wald chi2[3] = 31.71, p < .0001), and there were no consistent sex or race/ethnicity differences. Injection network size was smaller in the oldest (45+) age category (chi2[1] = 6.35, p = .01), and non-Hispanic white PWID tended to report a larger number of people they injected with compared to other categories (chi2[1] = 12.28, p = .0005). Homosexual men reported smaller PWID networks (IRR = 0.74, 95% CI 0.57–0.95), while bisexual women reported larger PWID networks (IRR = 1.17, 95% CI 1.07–1.29), and bisexual men and women reported injecting with a larger number of people (IRR = 1.14, 95% CI 1.02–1.27).

**Network mixing.** On average, men and women reported respectively that 68% (95% CI 0.67–0.70, tau = 0.01) and 60% (95% CI 0.58–0.64, tau = 0.03) of the people they knew who injected drugs were male. Young PWID (under 25 years old) reported on average 61% (95% CI 0.42–0.79, tau = 0.16) of their PWID network was also under 25, while older PWID reported 14% (95% CI 0.13–0.15, tau = 0) of their PWID network was under 25. Among PWID who lived in the city of Chicago, 87% (95% CI 0.79–0.94, tau = 0.06) of network members also lived in Chicago, while among PWID who lived in surrounding suburban areas or

**Table 4. Meta-analysis estimates of injection network size, by sex, race or ethnicity, and age category.**

| Sex | Age category | non-Hispanic white | | | non-Hispanic Black | | | Hispanic | | | Other non-Hispanic | | |
|---|---|---|---|---|---|---|---|---|---|---|---|---|---|
| | | Mean | LL | UL | Mean | LL | UL | Mean | LL | UL | Mean | LL | UL |
| Male | 18–24 | 4.13 | 3.35 | 4.90 | 8.00† | | | 3.62 | 2.63 | 4.61 | 3.26 | 1.08 | 5.44 |
| | 25–34 | 4.96 | 2.81 | 7.11 | 2.39 | 1.84 | 2.93 | 3.37 | 1.78 | 4.97 | 3.97 | 1.75 | 6.19 |
| | 35–44 | 6.28 | - - | - - | 5.10 | - - | - - | 4.91 | - - | - - | NA | - - | - - |
| | 45+ | 4.45 | - - | - - | 3.97 | - - | - - | 4.29 | - - | - - | 3.71† | - - | - - |
| Female | 18–25 | 4.08 | 3.15 | 5.01 | NA | | | 3.39 | 1.42 | 5.37 | 3.70 | 2.91 | 4.48 |
| | 25–34 | 3.57 | 2.67 | 4.48 | 2.00 | 1.02 | 2.99 | 3.60 | 2.24 | 4.97 | 6.44 | -2.99 | 15.86 |
| | 35–44 | 9.45 | - - | - - | 2.77 | - - | - - | 3.72 | - - | - - | NA | - - | - - |
| | 45+ | 4.20 | - - | - - | 3.80 | - - | - - | 2.71 | - - | - - | NA | - - | - - |

LL: lower limit 95% confidence interval; UL upper limit 95% confidence interval; NA: estimate not available, insufficient data.

† N < 10.

Note: confidence intervals not computed for estimates based on a single study.

nearby states, 37% (95% CI 0.19–0.55, tau = 0.16) of network members lived in Chicago. Non-Hispanic white PWID reported that 74% (95% CI 0.62–0.85, tau = 0.11) of their network members were also white, non-Hispanic Black PWID reported that 55% (95% CI 0.28–0.81, tau = 0.26) of their network members were also non-Hispanic Black, and Hispanic PWID reported that 51% (95% CI 0.33–0.68, tau = 0.18) of their network members were also Hispanic.

**Injection risk behavior.** Overall, the estimated percentage of PWID reporting equipment sharing was 63% (95% CI 0.56–0.69, tau = 0.39), receptive syringe sharing (RSS) was 31% (95% CI 0.25–0.38, tau = 0.55), distributive syringe sharing (DSS) was 39% (95% CI 0.35–0.44, tau = 0.22), and syringe mediated sharing was 25% (95% CI 0.22–0.28, tau = 0.28). The average proportion of injections involving RSS was 0.12 (95% CI 0.09–0.15, tau = 0.05), and the average proportion of injections involving shared equipment was 0.28 (95% CI 0.23–0.33, tau = 0.70). The results for proportions of PWID in each demographic subgroup reporting any receptive and distributive syringe sharing are shown in Figs 1 and 2. Additional results are available in S2 File. Of note, several subgroups were small leading to large confidence intervals for the estimates.

In mixed effects regressions, women were more likely than men to report equipment sharing (OR = 1.32, 95% CI 1.17–1.50), RSS (OR = 1.47, 95% CI 1.37–1.59) and DSS (OR = 1.38, 95% CI 1.26–1.52). There were also significant effects of age on all three behaviors. The likelihood of recent equipment sharing decreased with age (Chi2[3] = 10.80, p = .01), from 0.69 in the youngest (18–24) category to 0.61 and 0.58 in the intermediate age categories, and 0.56 in the oldest ($\geq$ 45) category. The likelihood of recent RSS also decreased with age (Chi2[3] = 15.18, p = .002), from 0.37 for the 18–24 category to 0.31 and 0.28 in the intermediate categories, and 0.23 for 45 and older. The likelihood of recent DSS was significantly lower for the

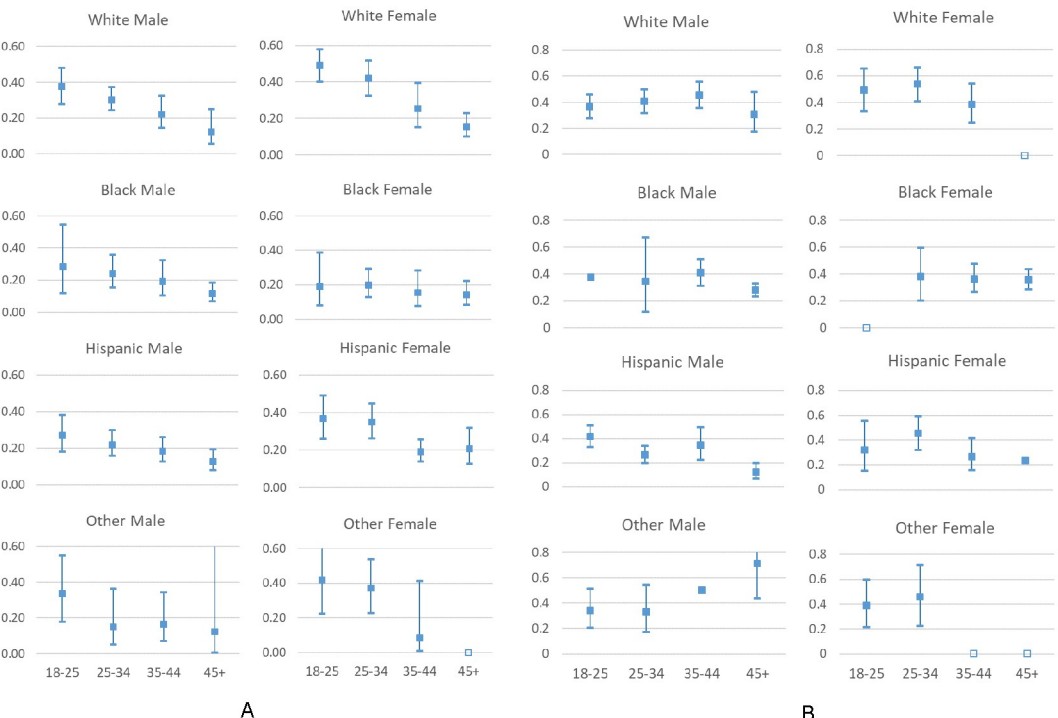

**Fig 1.** Proportions of PWID reporting (A) receptive syringe sharing. (B) distributive syringe sharing. Filled box indicates point estimate, whiskers indicate 95% confidence interval; empty box indicates insufficient data for estimate.

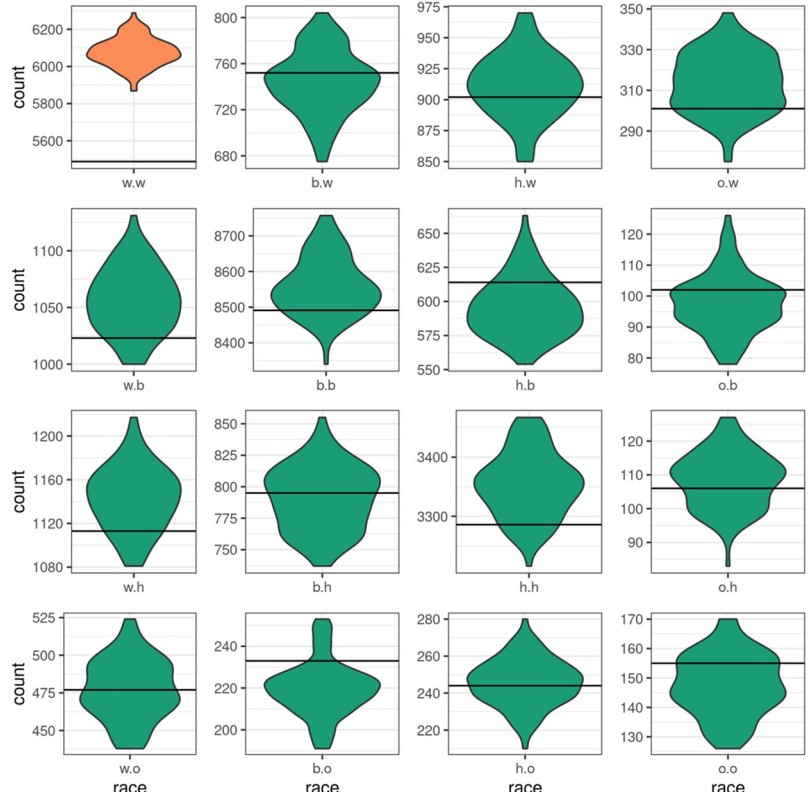

**Fig 2. Simulated and target statistics for race mixing.** The labels "w", "b", "h" and "o" represent White, Black, Hispanic and Other. The White-White term (shown in orange) was left out to avoid collinearity.

oldest age category (≥ 45) compared to all other categories (Chi2[1] = 31.02, p < .0001), dropping from 0.42 in the youngest category and 0.41 and 0.39 in the intermediate categories, to 0.29 in the oldest category. Hispanics were less likely to report equipment sharing compared to the other race/ethnicity categories (Chi2[1] = 13.5, p = .0002; Hispanic vs. white OR = 0.80, 95% CI 0.71–0.90), while RSS was more likely to be reported by white compared to Black or Hispanic PWID (Chi2[3] = 12.47, p = .006; Black vs. white OR = 0.75, 95% CI 0.56–1.01; Hispanic vs. white OR = 0.82, 95% CI 0.70–0.95). The likelihood of DSS was not influenced by race/ethnicity. Persons residing in the city of Chicago were less likely to report RSS (29% vs. 34%, OR = 0.79, 95% CI 0.67–0.92) compared to those living outside of Chicago, even while adjusting for other variables. Homosexual (OR = 1.87, 95% CI 1.21–2.89) and bisexual (OR = 2.14, 95% CI 1.71–2.66) PWID were more likely to report RSS (vs. heterosexual), bisexual PWID were more likely to report equipment sharing (OR = 1.74, 95% CI 1.48–2.04), and bisexual men (OR = 1.85, 95% CI 1.36–2.52) and homosexual (OR = 3.12, 95% CI 1.58–6.16) and bisexual women (OR = 1.65, 95% CI 1.25–2.18) were more likely to report DSS.

**Sharing partners.** Among PWID who shared any injection equipment, the estimated average number of sharing partners was 3.5 (95% CI 2.6–4.4, tau = 1.13). There were no significant effects of demographic variables in negative binomial mixed effects regressions on equipment sharing partners. The estimates of number of receptive and distributive syringe sharing partners are shown in Tables 5 and 6. Among PWID who used a shared syringe, the overall estimated average number of syringe-sharing partners was 2.2 (95% CI 1.64–2.67, tau = 0.78).

**Table 5. Meta-analysis estimates of number of receptive syringe-sharing (RSS) injection partners among PWID reporting recent RSS, by sex, race or ethnicity, and age category.**

| Sex | Age category | non-Hispanic white | | | non-Hispanic Black | | | Hispanic | | | Other non-Hispanic | | |
|---|---|---|---|---|---|---|---|---|---|---|---|---|---|
| | | Mean | LL | UL | Mean | LL | UL | Mean | LL | UL | Mean | LL | UL |
| Male | 18–24 | 1.61 | 1.37 | 1.85 | 8.00[†] | | | 2.28 | 1.12 | 3.44 | 1.60 | - | - |
| | 25–34 | 1.91 | 1.31 | 2.52 | 1.22 | 0.62 | 1.82 | 1.84 | 1.42 | 2.27 | NA | | |
| | 35–44 | 1.97 | 0.97 | 2.98 | 2.15 | 1.30 | 2.99 | 2.59 | 1.37 | 3.81 | NA | | |
| | 45+ | 3.32 | 1.68 | 4.96 | 3.65 | 1.23 | 6.07 | 2.37 | 0.99 | 3.75 | NA | | |
| Female | 18–25 | 1.58 | 1.34 | 1.81 | NA | | | 1.82 | 1.15 | 2.48 | 1.78 | - | - |
| | 25–34 | 2.06 | 1.29 | 2.82 | 1.00 | - | - | 2.22 | 0.97 | 3.47 | NA | | |
| | 35–44 | 1.83 | 0.53 | 3.13 | 3.31 | 2.14 | 4.48 | 1.53 | 0.51 | 2.55 | NA | | |
| | 45+ | 1.05 | 0.55 | 1.56 | 1.78 | 0.93 | 2.63 | NA | | | NA | | |

LL: lower limit 95% confidence interval; UL: upper limit 95% confidence interval; NA: estimate not available, insufficient data.

[†] N < 10.

Note: confidence intervals not computed for estimates based on a single study.

Among PWID who gave a used syringe to another person to use, the estimated average number of people shared with was 2.4 (95% CI 1.86–2.98, tau = 0.62).

In mixed effects negative binomial regressions, there were significant effects of age on number of RSS partners (Chi2[3] = 12.92, p = .005) and DSS partners (Chi2[3] = 18.88, p = .0003). The marginal mean numbers of RSS and DSS partners were 2.1 and 2.2, respectively in the youngest age category, and 2.5 and 2.6, respectively in the oldest age category. There was also a significant effect of race/ethnicity on number of DSS partners (Chi2[3] = 27.00, p < .0001), with non-Hispanic other race/ethnicity reporting fewer DSS partners (marginal mean 2.0; vs. white IRR = 0.82, 95% CI 0.72–0.94). There was no difference in number of RSS or DSS partners by sex or urban residence.

## Fitting and simulating synthetic syringe sharing networks

**Model fit.** We define the random edge probability as the number of observed edges divided by the maximum number of possible edges on a directed network with 32,000 nodes.

**Table 6. Meta-analysis estimates of number of distributive syringe sharing (DSS) injection partners among PWID reporting recent DSS, by sex, race or ethnicity, and age category.**

| Sex | Age category | non-Hispanic white | | | non-Hispanic Black | | | Hispanic | | | Other non-Hispanic | | |
|---|---|---|---|---|---|---|---|---|---|---|---|---|---|
| | | Mean | LL | UL | Mean | LL | UL | Mean | LL | UL | Mean | LL | UL |
| Male | 18–24 | 1.83 | 1.51 | 2.14 | NA | | | 2.10 | 1.50 | 2.70 | 1.81 | 1.05 | 2.57 |
| | 25–34 | 2.62 | 1.72 | 3.53 | 1.80 | 0.97 | 2.63 | 2.11 | 1.66 | 2.56 | 1.80 | - | - |
| | 35–44 | 4.60 | 2.33 | 6.86 | 2.84 | 2.31 | 3.38 | 3.37 | 1.85 | 4.90 | NA | | |
| | 45+ | 3.41 | 1.63 | 5.19 | 3.29 | 2.51 | 4.07 | 3.30 | - | - | 3.00 | - | - |
| Female | 18–25 | 1.90 | 1.54 | 2.26 | NA | | | 3.06 | - | - | 2.00 | - | - |
| | 25–34 | 2.81 | 1.69 | 3.93 | NA | | | 2.52 | 1.28 | 3.77 | NA | | |
| | 35–44 | 2.01 | 1.29 | 2.73 | 3.04 | - | - | 2.73 | - | - | NA | | |
| | 45+ | NA | | | 2.97 | 1.60 | 4.33 | NA | | | NA | | |

LL: lower limit 95% confidence interval; UL: upper limit 95% confidence interval; NA: estimate not available, insufficient data.

[†] N < 10.

Note: confidence intervals not computed for estimates based on a single study.

We report the ratio ($R_p$) of the probability of the edge corresponding to a specified network parameter relative to a random edge. Estimated $R_p$ for edges representing needles shared from males to females is 2.41, from females to males is 2.50, and between males is 1.43, indicating the higher probability of syringe sharing from males to females, and between females than between males. The $R_p$ values for mixing between age categories are: 5.98 for persons ≤25 years of age; 1.98 for edges representing syringes shared from persons ≤25 years to persons >26 years and 0.74 for edges representing syringe sharing from persons >26 years to persons ≤25 years. We also note that syringe sharing shows strong race homophily as evidenced by the $R_p$ values for edges between Black (9.07), Hispanic (5.57), and all other race/ethnicities (9.86) relative to the much lower $R_p$ values representing edges between different races (Table A.1 in S1 File). Syringe sharing between PWID (nodes) residing less than 1/8 mile or 1/8–1 mile apart are much more likely ($R_p$ values of 1511 and 58.13, respectively); $R_p$ for an edge between nodes 1–20 miles apart is about 0.8. See S1 File for a detailed summary of the estimated model coefficients and relative probabilities.

**Comparing the fit of the simulated models to the targets.**   The distributions of the specified network parameters across the 100 simulated networks and the target values for each of the statistics is shown in Table 7. The target values of each of the network parameters specified in the model (see S1 File) are within the 2.5th and 97.5th percentiles of the simulated distribution.

## Discussion

Our meta-analysis of the last 20 years of data on people who inject drugs provide a profile of this population from a large metropolitan area. To our knowledge, this is the first meta-analysis of recent PWID empirical data that more realistically represent the demographic and geographic shift in the PWID population composition resulting from on-going opioid epidemic [47].

As we would expect, PWID have high rates of joblessness and homelessness. In a previous paper, we analyzed a subset of these data and found that homelessness among young PWID increased significantly over the covered time period [32]. Young white suburban male PWID are somewhat better off than other subgroups. This might in part explain why young male PWID are less likely to obtain syringes from a SSP, as they may be able to purchase them at drug stores more conveniently. However, this means they are less likely to be exposed to harm reduction services such as syringe exchange, MAT, HIV/HCV testing and counseling, and drug and HIV/HCV treatment referrals.

Older PWID were both more likely to use a SSP and less likely to report sharing of syringes and other equipment. However, older PWID who did share syringes tended to have more sharing partners than their younger counterparts, which could be an indication of higher HIV and HCV infection risk for this group. In contrast to age effects, the findings for women were inconsistent. Although women were more likely than men to use a SSP, they were also more likely to report sharing of syringes and other equipment. Minority sexual orientation was also associated with larger risk networks and greater likelihood of syringe sharing among both men and women. PWID living in Chicago were less likely to report RSS, perhaps reflecting greater access to SSPs. Yet, in spite of greater availability of SSPs in Chicago, non-Hispanic Black PWID were less likely than others of the same age to use a SSP. This highlights the need for greater outreach efforts to reach Black urban PWID who are at high risk for HCV infection and opioid overdose.

ERGMs allow for fitting of a model that incorporates mixing and degree parameters that describe processes that govern formation of syringe sharing networks in a statistically robust

**Table 7. Simulated network parameters and target statistics.**

| Parameter | Target Statistic | Mean (2.5, 97.5 percentiles)[a] |
|---|---|---|
| Number of edges | 24650 | 24840 (24504, 25149) |
| Number of nodes with out-degree: | | |
| Degree 0 | 20376 | 20315 (20167, 20451) |
| Degree 1 | 5368 | 5382 (5245, 5485) |
| Degree 2 | 2578 | 2590 (2497, 2685) |
| Degree 3 | 1424 | 1429 (1365, 1494) |
| Degree > 3 [b] | 2254 | 2284 (2238, 2343) |
| Number of nodes with in-degree: | | |
| Degree 0 | 23162 | 23117 (22991, 23264) |
| Degree 1 | 4159 | 4178 (4079, 4281) |
| Degree > 1 [b] | 4679 | 4705 (4636, 4773) |
| Race mixing: | | |
| Black-White | 752 | 744 (688, 791) |
| Hispanic-White | 902 | 911 (854, 964) |
| Other-White | 301 | 313 (291, 339) |
| White-Black | 1023 | 1056 (1002, 1107) |
| Black-Black | 8491 | 8559 (8438, 8727) |
| Hispanic-Black | 614 | 596 (561, 643) |
| Other-Black | 102 | 98 (80, 119) |
| White-Hispanic | 1113 | 1143 (1085, 1198) |
| Black-Hispanic | 795 | 791 (747, 838) |
| Hispanic-Hispanic | 3286 | 3353 (3259, 3460) |
| Other-Hispanic | 106 | 108 (94, 125) |
| White-Other | 477 | 477 (439, 517) |
| Black-Other | 233 | 219 (193, 250) |
| Hispanic-Other | 244 | 245 (222, 269) |
| Other-Other | 155 | 148 (129, 170) |
| White-White [b] | 5488 | 6077 (5941, 6222) |
| Gender-mixing: | | |
| Female-Male | 7487 | 7580 (7463, 7750) |
| Male-Female | 6784 | 6770 (6597, 6926) |
| Male-Male | 7674 | 7694 (7532, 7843) |
| Female-Female [b] | 2699 | 2797 (2715, 2887) |
| Age-mixing | | |
| Young-Old | 3170 | 3132 (3037, 3223) |
| Old-Young | 1008 | 990 (935, 1056) |
| Old-Old | 1474 | 1488 (1429, 1556) |
| Young-Young [b] | 18992 | 19230 (18949, 19487) |
| Distance (miles): | | |
| Category 1 (< 1/8) | 3870 | 3970 (3948, 3995) |
| Category 2 (1/8–1) | 8652 | 8722 (8609, 8835) |
| Category 3 (1–20) | 5940 | 5980 (5827, 6131) |
| Category 4[b] (>20) | 5423 | 6168 (5943, 6354) |

[a] across 100 simulated networks

[b] category omitted for estimation.

fashion. In this study, we estimated the important impacts of network characteristics such as sex, age, race and ethnicity, and geographic proximity, in addition to the network processes that govern the process of sharing and receiving a syringe (i.e, out- and in-ties, respectively). We found that syringe sharing from males to females and between females was more likely than syringe sharing between males. We also estimated the impact of geographic proximity and race-based homophily on the syringe sharing in the population.

## Limitations

Our study has notable limitations associated with the methods used as well as the sources of data. When study parameters are heterogeneous, the pooled estimate alone is an insufficient summary of the underlying population parameter. It is also important to understand the variability in the estimates. We conducted mixed effects regression analyses to inform our understanding of the sources of variability among the studies. However, there are likely additional sources of variability beyond the demographic characteristics that we investigated. While measures were harmonized to be as equivalent as possible across studies, differences across studies in time frame of measurement and definitions may have contributed to between study heterogeneity of the meta-analysis estimates. Second, we applied the common assumption of a normal distribution for the random effects. Alternative methods that do not rely on this assumption might produce different estimates.

The datasets represent large, robust observational studies and intervention program records of hard-to-reach and retain individuals. A shortcoming inherent in the study of hard-to-reach populations is that the representativeness of the samples is unknown. We did not have a large number of data sources, and for some measures, only a subset of these provided data. Participants were predominantly non-Hispanic white and samples contained small proportions of Black and Hispanic PWID, limiting subgroup analyses. In computing estimates for demographic subgroups, the data available were sometimes insufficient to produce an estimate. An alternative approach could be to use marginal predicted values from mixed effects models, that would essentially smooth over the holes in the data.

In spite of these limitations, the meta-analysis estimates provide an empirical basis for setting parameters such as average injection network size and the probability of syringe sharing in a synthetic population for complex computational models, including ABMs developed by our group [29, 48]. Our findings may not be generalizable to PWID in other geographic locations or settings, but rather represent a starting point for other regions.

## Conclusions

Our meta-analysis of data from multiple diverse datasets on PWID, and the fitted ERGM model using the network estimates from the meta-analysis, generated robust estimates on a circumscribed population during the on-going opioid epidemic, which can provide a useful resource for developing intervention strategies in this population. Computational models allow for the discovery of interventions or combinations of interventions prior to implementation that accounts for the complex interplay of multilevel factors. A direct application of this study's findings could be to provide the estimates for generating a data-driven, realistic synthetic population for complex computational models that build on on-going efforts in Chicago [48]. In turn, these models might be used model the effectiveness of strategies for HCV microelimination and/or other outcomes (e.g. overdose prevention) for Chicago, which can be adapted for other regions. Future work will include building agent-based models.

## Supporting information

**S1 File. Technical appendix.** Model fitting procedure, code, and table of estimated coefficients, standard errors and mean edge probability and probability ratio obtained from exponential random graph model.
(PDF)

**S2 File. S1-S3 Figs.** Figures showing estimates for proportions of PWID reporting sharing of ancillary injection equipment (S1 Fig), syringe-mediated drug sharing (S2 Fig), and obtaining syringes from a syringe service program (S3 Fig).
(PDF)

**S1 Table. Random effects meta-analysis estimates of characteristics of PWID population.**
(PDF)

## Acknowledgments

We thank Dr. Lawrence Ouellet and the Chicago Department of Public Health for providing access to several data sets, and we extend our appreciation to Community Outreach Intervention Projects for providing access to service program data.

## Author Contributions

**Conceptualization:** Basmattee Boodram, Mary Ellen Mackesy-Amiti, Jonathan Ozik.

**Data curation:** Mary Ellen Mackesy-Amiti.

**Formal analysis:** Mary Ellen Mackesy-Amiti.

**Funding acquisition:** Basmattee Boodram, Harel Dahari.

**Methodology:** Mary Ellen Mackesy-Amiti, Aditya Khanna, Jonathan Ozik.

**Software:** Aditya Khanna, Bryan Brickman, Jonathan Ozik.

**Visualization:** Mary Ellen Mackesy-Amiti, Bryan Brickman.

**Writing – original draft:** Basmattee Boodram, Mary Ellen Mackesy-Amiti.

**Writing – review & editing:** Basmattee Boodram, Mary Ellen Mackesy-Amiti, Aditya Khanna, Harel Dahari, Jonathan Ozik.

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
