## [Decision Letter · Decision Letter 0]

3 Jun 2021

PONE-D-21-09358

A meta-analysis of 20 years of data on people who inject drugs in metropolitan Chicago to inform computational modeling

PLOS ONE

Dear Dr. Mackesy-Amiti,

Thank you for submitting your manuscript to PLOS ONE. After careful consideration, we feel that it has merit but does not fully meet PLOS ONE’s publication criteria as it currently stands. Therefore, we invite you to submit a revised version of the manuscript that addresses the points raised during the review process.

We look forward to receiving your revised manuscript.

Kind regards,

Ngai Sze Wong

Academic Editor

PLOS ONE

Additional Editor Comments:

When responding to the reviewer's concerns, it will be particularly important to respond with significant changes (adding limitations in DISCUSSION and assessment of heterogeneity in methods and results), and clarification to the concern of drug types (or all were heroin users?).

The DISCUSSION part usually includes comparison of results with other studies, and implications of methods or results. Also, there is no in-text citation for 'Table 3'.

Journal Requirements:

2. We note you have included a table to which you do not refer in the text of your manuscript. Please ensure that you refer to Table 3 in your text; if accepted, production will need this reference to link the reader to the Table.

Reviewers' comments:

Reviewer's Responses to Questions

**Comments to the Author**

1. Is the manuscript technically sound, and do the data support the conclusions?

Reviewer #1: No

2. Has the statistical analysis been performed appropriately and rigorously? 

Reviewer #1: Yes

3. Have the authors made all data underlying the findings in their manuscript fully available?

Reviewer #1: Yes

4. Is the manuscript presented in an intelligible fashion and written in standard English?

Reviewer #1: Yes

5. Review Comments to the Author

Reviewer #1: Thank you for the opportunity to review this interesting manuscript. While it is impressive in some respects, I have some major concerns regarding some data, analytic and interpretable omissions that limit my enthusiasm.

While I agree with the authors that a random-effects meta-analysis is appropriate given potential heterogeneity in the data being analyzed, this does not remove issues of heterogeneity. To fully interpret these findings, tests of heterogeneity are still required (I would suggest I2 as well as Heterogeneity Variance τ2 /Standard Deviation τ to produce a standard deviation of the true effect sizes). It would also be helpful to inform readers about the limits of interpretability related to the use of random effects meta-analysis.

To that end, there is no limitations section, which is of concern given that there are multiple complex analytic approaches presented in this manuscript, and the data are generated from prospective observational cohort studies of hard-to-reach and retain individuals. The limitations of this work are many and important, and should be clearly elucidated.

One critical piece not included is the type of drug injected. This is highly determinative for syringe related risk behaviors and, to a certain extent, network size. These data need to be incorporated into the analysis, even if broadly (i.e., by drug type: opioid, amphetamine-type stimulant, other stimulants [cocaine/crack cocaine], and tranquilizers). Not including these data is a critical flaw.

Broadly (and this is explicated more specifically below), there are some important contextual features from the studies—mostly around when they were conducted and the inclusion of gender and sexual minorities—that could be critically helpful in understanding what drives network size and syringe-related risk behaviors. At minimum, understanding whether there were time trends in the reporting of certain outcomes (i.e., the average proportion of syringe sharing, injection frequency, accessing SSP) is needed to contextualize the findings.

Further comments:

What was the assumed distribution (normal, etc.) employed in the meta-analysis?

Did associations or trends change over time? For example, were there changes in the median income level, proportion of homeless respondents, proportion of Hispanic or Black respondents, age of first injection, use of SSP, accessing treatment, network size, etc.? These would be really helpful data to present.

Was median income inflation-adjusted and if so, what was the reference year?

Sexual identity appears critical in influencing risk and in determining the number of out-network ties; were details on sexual identity/preference collected? Even if they were for a subset of the studies I would strongly suggest running a subanalysis on those as this is likely to be determinative for syringe-related risk behaviors. For instance, young gay men who inject have in many settings be identified as particularly high risk for injection-related risk behaviors, mediated in part by use of stimulants (e.g., crystal methamphetamine).

Though cell sizes may be small, it would also be important if possible to explore gender identities in relation to risk, even if limited to descriptive statistics.

Unclear about the proportion of equipment sharing: is it 62% or 0.62%? If the former, which seems more reasonable, there seems to be some inconsistent presentation of values here.

Unfortunate that young-young mixing was not estimable. What is the implication for the strengths of this meta-analysis to model network size?

The discussion really does not delve into the issues raised by the results at all. It is largely a summary of the results.

6. PLOS authors have the option to publish the peer review history of their article (what does this mean?). If published, this will include your full peer review and any attached files.

Reviewer #1: No

---

## [Author Response · Author response to Decision Letter 0]

11 Oct 2021

We thank the reviewers for their insightful comments. We have addressed all of the comments as follows below. In addition to the requested revisions, we have edited the entire manuscript to enhance clarity and flow, particularly in the introduction and discussion. We have also elected to revise the manuscript title to reflect these changes. 

1. While I agree with the authors that a random-effects meta-analysis is appropriate given potential heterogeneity in the data being analyzed, this does not remove issues of heterogeneity. To fully interpret these findings, tests of heterogeneity are still required (I would suggest I2 as well as Heterogeneity Variance τ2 /Standard Deviation τ to produce a standard deviation of the true effect sizes)..

Response: Due to some concerns about the Freeman-Tukey method for proportions†, we re-ran the analysis in R using the GLMM method to obtain measures of heterogeneity including τ2 and I2. Rather than using different software for some measures, we decided to re-run all analyses using the R meta procedure. This resulted in some slight changes in the estimates, particularly for confidence intervals. The R procedure does not generate confidence intervals for estimates based on only one study. We now also present values of tau in the text. We also corrected an oversight in the analysis of numbers of RSS and DSS partners in which study estimates based on small cell sizes (<5) were not excluded. 

†Schwarzer G, Chemaitelly H, Abu-Raddad LJ, Rücker G. Seriously misleading results using inverse of Freeman-Tukey double arcsine transformation in meta-analysis of single proportions. Research Synthesis Methods. 2019;10(3):476-83. doi: https://doi.org/10.1002/jrsm.1348.

2. It would also be helpful to inform readers about the limits of interpretability related to the use of random effects meta-analysis. To that end, there is no limitations section, which is of concern given that there are multiple complex analytic approaches presented in this manuscript, and the data are generated from prospective observational cohort studies of hard-to-reach and retain individuals. The limitations of this work are many and important, and should be clearly elucidated.

Response: We thank the reviewers for this insight. We have added a limitations section to the revised manuscript that addresses these issues (see Discussion, Limitations). While we agree on the importance of quantifying variability in the estimates, since we are using meta-analysis to generate estimates for a synthetic population (rather than studying a treatment or intervention effect), heterogeneity is not a major limitation. 

3. One critical piece not included is the type of drug injected. 

Response: Over all studies included in the analysis, 86-100% of participants injected heroin by itself (pooled = 97%). Cocaine was the second most common drug injected (pooled = 36%), either alone or with heroin (speedball). We have clarified this in the revised manuscript (see Methods, Meta-analysis Data Sources section). 

4. What was the assumed distribution (normal, etc.) employed in the meta-analysis?

Response: The assumed distribution of random effects was normal. We have clarified this in revised manuscript (see Methods, Meta-analysis: Random Effects Model section)

5. Understanding whether there were time trends in the reporting of certain outcomes (i.e., the average proportion of syringe sharing, injection frequency, accessing SSP) is needed to contextualize the findings. Did associations or trends change over time? For example, were there changes in the median income level, proportion of homeless respondents, proportion of Hispanic or Black respondents, age of first injection, use of SSP, accessing treatment, network size, etc.? These would be really helpful data to present.

Response: Due to confounding of study with time, analysis of time trends is limited to using meta-regression on aggregate level data (rather than mixed effects regression). In another paper (Hotton et al, 2021 listed below), we report an increase in homelessness over time among young PWID. We detected no significant changes in risk behavior or networks over time. The demographic composition of the samples is not necessarily a good indicator of the evolving demographics of the metro PWID population, so we did not try to estimate changes in racial/ethnic composition for example. (We used other sources and methods to construct the demographic profile for the ABM.) We have included an explanation of this in the revised manuscript (see Methods, Meta-analysis: Random Effects Model section). 

Hotton AL, Mackesy-Amiti ME, Boodram B. Trends in homelessness and injection practices among young urban and suburban people who inject drugs: 1997-2017. Drug Alcohol Depend. 2021;225:108797. Epub 2021/06/09. doi: 10.1016/j.drugalcdep.2021.108797. 

6. Was median income inflation-adjusted and if so, what was the reference year?

Response: Median income was not inflation-adjusted, since we were not looking at time trends, we were only testing associations between income and other demographic variables. As such, no revisions to manuscript was made in this regard. 

7. Sexual identity appears critical in influencing risk and in determining the number of out-network ties; were details on sexual identity/preference collected? Even if they were for a subset of the studies I would strongly suggest running a subanalysis on those as this is likely to be determinative for syringe-related risk behaviors. For instance, young gay men who inject have in many settings be identified as particularly high risk for injection-related risk behaviors, mediated in part by use of stimulants (e.g., crystal methamphetamine). Though cell sizes may be small, it would also be important if possible to explore gender identities in relation to risk, even if limited to descriptive statistics.

Response: Sexual orientation was collected in 9 of the 11 studies, and transgender identity was collected in all studies. Very few trans-identified persons were included in these samples (n=22). There were 114 respondents who identified as gay/homosexual (pooled estimate 2.9%), and 796 who identified as bisexual (pooled estimate 10.8%). In the revised manuscript, we conducted additional mixed effects regression analyses to examine the contribution of sexual orientation to risk behavior and network size (See Methods and Results sections). 

8. Unclear about the proportion of equipment sharing: is it 62% or 0.62%? If the former, which seems more reasonable, there seems to be some inconsistent presentation of values here.

Response: The percentage reporting use of a shared syringe was 62%. We have fixed this inconsistency in the revised manuscript.

9. Unfortunate that young-young mixing was not estimable. What is the implication for the strengths of this meta-analysis to model network size?

Response: In estimating mixing using ERGMs, a base term needs to be specified. In this case, we selected the young-young term as the base. The log-odds estimated for the remaining mixing combinations (young-old, old-old, old-young) are estimated relative to this base term. We see from Table 1 that this specification works well; the target number of young-young edges is 18992, which falls within the IQR produced across our 100 simulations: (18949, 19487). The target statistics for the remaining combinations of age mixing similarly fall within the IQRs of the relevant statistics across the simulated networks. Given that these reference groups are already specified, not revisions to the manuscript is made in regard to this. 

More details on the specification of mixing terms in ERGMs can be found in Goodreau et. al. below. We have included this clarification in the methods section pertaining to ERGMs.

Goodreau, S. M., Handcock, M. S., Hunter, D. R., Butts, C. T., & Morris, M. (2008). A statnet Tutorial. Journal of Statistical Software, 24(9), 1–27.

10. The discussion really does not delve into the issues raised by the results at all. It is largely a summary of the results.

Response: We have substantially expanded discussion of the results in the revised manuscript. In addition, we have provided additional literature the introduction to support the premise of the study, and therefore, the utility of the results in its applications.

---

## [Editor Report · Decision Letter 1]

14 Dec 2021

People who inject drugs in metropolitan Chicago: A meta-analysis of data from 1997-2017 to inform interventions and computational modeling toward hepatitis C microelimination

PONE-D-21-09358R1

Dear Dr. Mackesy-Amiti,

We’re pleased to inform you that your manuscript has been judged scientifically suitable for publication and will be formally accepted for publication once it meets all outstanding technical requirements.

Kind regards,

Ngai Sze Wong

Academic Editor

PLOS ONE

---

## [Editor Report · Acceptance letter]

3 Jan 2022

PONE-D-21-09358R1 

People who inject drugs in metropolitan Chicago: A meta-analysis of data from 1997-2017 to inform interventions and computational modeling toward hepatitis C microelimination 

Dear Dr. Mackesy-Amiti:

I'm pleased to inform you that your manuscript has been deemed suitable for publication in PLOS ONE. Congratulations! Your manuscript is now with our production department. 

Kind regards, 

on behalf of

Dr. Ngai Sze Wong 

Academic Editor

PLOS ONE